# Concepts and Core Principles of Fragment-Based Drug Design

**DOI:** 10.3390/molecules24234309

**Published:** 2019-11-26

**Authors:** Philine Kirsch, Alwin M. Hartman, Anna K. H. Hirsch, Martin Empting

**Affiliations:** 1Helmholtz-Institute for Pharmaceutical Research Saarland (HIPS)-Helmholtz Centre for Infection Research (HZI), Department of Drug Design and Optimization (DDOP), Campus E8.1, 66123 Saarbrücken, Germany; Philine.Kirsch@helmholtz-hzi.de (P.K.); Alwin.Hartman@helmholtz-hzi.de (A.M.H.); Anna.Hirsch@helmholtz-hzi.de (A.K.H.H.); 2Department of Pharmacy, Saarland University, Campus E8.1, 66123 Saarbrücken, Germany; 3German Centre for Infection Research (DZIF), Partner Site Hannover-Braunschweig, 66123 Saarbrücken, Germany; 4Stratingh Institute for Chemistry, University of Groningen, Nijenborgh 7, 9747 AG Groningen, The Netherlands

**Keywords:** fragment-based drug design, biophysical screening, rule-of-three, ligand efficiency, fragment optimization

## Abstract

In this review, a general introduction to fragment-based drug design and the underlying concepts is given. General considerations and methodologies ranging from library selection/construction over biophysical screening and evaluation methods to in-depth hit qualification and subsequent optimization strategies are discussed. These principles can be generally applied to most classes of drug targets. The examples given for fragment growing, merging, and linking strategies at the end of the review are set in the fields of enzyme-inhibitor design and macromolecule–macromolecule interaction inhibition. Building upon the foundation of fragment-based drug discovery (FBDD) and its methodologies, we also highlight a few new trends in FBDD.

## 1. Introduction

Fragment-based drug discovery (FBDD) has become of increasing importance and interest in the past decades, especially in academia [1]. FBDD uses the advantages of biophysical and biochemical methods for the detection of very small molecules or so-called “fragments” binding to a specific target. If selected by thorough evaluation, fragment hits provide fascinating and facile starting points for the generation of drug leads. Initial identification can be achieved by the application of an array of different biophysical methods, which we discuss in the main part of our review [2]. Typically, FBDD starts with a screening of a small library of low molecular weight compounds for binding to a particular target. The key advantage of fragments is their low degree of complexity. Actually, these scaffolds should hit a sweet spot, which enables them to still be big enough to undergo a few directed attractive interactions with the protein of interest so that biophysical detection of target binding is possible. On the other hand, they should retain a sufficiently small size to limit the danger of unfavourable interactions, such as steric clashes [3]. Hence, fragments that are identified usually represent ideal binding motifs as desirable starting points for further optimization. The initial fragment hits generally have a weak binding affinity to their target, usually in a µM–mM range [4]. This can be explained by the fact that fragments possess fewer heavy atoms able to form multiple attractive interactions with the surface of the protein in comparison to larger molecules [5]. Additionally, fragments are small enough for binding to regions, which are often hard to target; for example; allosteric sites or essential small binding pockets termed hot spots, which are often central in protein-protein interactions. Enhancing the potency of the fragments after hit identification can be achieved by fragment linking, merging or growing strategies, which are key approaches for fragment-based drug design. The benefits of FBDD compared to the traditional high-throughput screening (HTS) have been well documented, rendering the former as a valuable alternative method in drug discovery. The main differences between FBDD and HTS are the specific compositions and the sizes of the libraries, and the assay methods used for hit identification. Fragment-based biophysical screening techniques require a high degree of sensitivity in order to detect weak binders, whereas HTS assays need to be more specific to exclude false positives [6]. The ligand efficiencies of small ligands are usually higher than for larger ligands, and generation of lead compounds starting from fragment hits is primed to result in improved physicochemical properties [7].

## 2. Library Construction, Preselection, and General Considerations

### 2.1. General Principles for Library Design

Fragment-based drug discovery usually starts with screening of a relatively small compound library comprised of compounds with low molecular weights, up to 300 Da, called fragments [8]. The library compounds should be highly structurally diverse so that 500–1000 congeners are sufficient for sampling a large structural space. Typically, fragments have fewer than 20 heavy atoms and low molecular complexity [8,9]. In a fragment-based approach, the throughput compared to HTS is rather low, which is why, the selection and construction of the compound library should be carefully considered to generate high-quality hits. Additionally, it is essential to take the purpose of a library into account and deliberately select those dedicated to specific target classes; for instance, using a focused library for matrix-metalloproteinases (MMPs). To assemble a project-focused fragment library with a high chemical and structural diversity, there are some criteria which are important to consider. The outcome of an FBDD project is directly influenced by the composition of the library being applied [10]. Firstly, commercially available fragments and fragment libraries should be analysed. It is important to use a library that meets some primary criteria depending on the profile of the respective target [11]. Usually, commercially available fragment libraries have been selected based on chemical and size diversity, and different, well-balanced properties to cover most of the important features. The overall diversity of the library can also be improved by using a pharmacophore-based selection. Secondly, a set of natural products or natural-product-inspired fragments are often useful and could be included [12]. Additionally, for future plans it is important to identify a series of non-commercially available fragments, which came from synthetic chemistry efforts; for example, from an in-house library or collaborating groups. Such scaffolds can provide a basis for future medicinal chemistry optimization strategies [13,14].

However, it must be noted that since these fragments are small, they may bind multiple targets, rendering them less sensitive to target classes. Selectivity can be generated in the course of fragment optimization [15]. Fragments should ideally hit the sweet spot between low molecular weight and still being big enough to realize specific interactions to a target [9]. Usually, fragments are more hydrophilic than compounds found in HTS libraries. Hence, the likeliness of specific hydrogen bonding is increased, resulting in favorable enthalpy-driven binding [16]. The small size of the compounds usually provides a suitable starting point and ample opportunities for a medicinal chemist to embark on optimization efforts facilitating the generation of drug-like molecules.

### 2.2. The Rule of Three

Typically, fragments obey the rule-of-three (RO3), requiring the following properties [5,17]:
The molecular weight is ≤300 Da;The number of hydrogen-bond donors ≤3;Number of hydrogen-bond acceptors ≤3;The logP is ≤3.


Furthermore, a number of rotatable bonds (NROT) ≤3 and a polar surface area (PSA) ≤60 can be beneficial [18]. The rule-of-three is closely related to the prominent Lipinski′s rule of five (RO5), which makes use of the same molecular descriptors but using a less restricted value of five as the cutoff (molecular weight ≤500 Da, number of hydrogen bond donors ≤5, number of hydrogen bond acceptors ≤10, logP ≤ 5). Importantly, the latter was derived to provide a predictor for the oral bioavailability of a drug candidate at hand, while the former is used to evaluate fragments regarding their suitability to be optimized into RO5-obeying drug candidates. RO3 as well as RO5, should be considered as rough guidelines in drug design for achieving orally bioavailable compounds. Importantly, it has to be noted that eventual drugs and clinical candidates can still have good bioavailability, whilst violating the Lipinski rules [19]. In line with this notion, library design trends that differ significantly to that suggested by the RO3 are being more frequently applied, as was reported by Pickett and coworkers [20].

### 2.3. Application of Electrophilic Fragments

In 2016, Backus et al. have described the use of a chemical proteomic method in order to perform fragment-based ligand discovery in native biological systems [21]. The authors report on the quantitative analysis of cysteine-reactive, small-molecule fragments that were screened against human proteomes and cells. Lysate as well as intact cells were treated with DMSO or an electrophilic fragment in a first step. In a second step, a cysteine-reactive probe containing an alkyne functional group was attached through copper-mediated, azide–alkyne cycloaddition with an azide-biotin tag. It was found that the druggability of the human proteome is larger than previously known, by expanding the search to proteins which were not known to interact with small molecules. More importantly, the applicability of reactive electrophilic fragments for the identification of covalent target binders has been demonstrated in further studies [22].

## 3. Fragment Hit Identification

The main challenge is to detect and select those fragments which are specifically binding to the target of interest. There are some tasks necessary for a successful FBDD campaign, such as carefully selecting the fragment library, applying several orthogonal methods to confirm that the fragment is binding to the target, and characterizing of the fragment’s binding mode. In this regard, orthogonal screening methods refer to, e.g., biophysical or biochemical techniques, which interrogate the ligand–target binding but rely on different measurement principles. This enables identification of the most promising fragment, which will be used for the generation of a lead compound [3]. Starting a fragment-screening campaign requires a robust screening cascade consisting of different biophysical binding assays to detect and confirm fragment binding [23]. Because of the low affinity of fragments, usually, high concentrations up to 1 or 2 mM are used for the primary selection. Due to the fact, that in a fragment screening the rate of non-specific binding and detection of false positives is particularly high, at least one orthogonal secondary screening method is required for hit confirmation [3]. After the primary screen with direct binding techniques, usually, a number of fragment hits have been identified as target binders. Afterwards, qualified hits can be optimized by various medicinal chemistry approaches and the new derivatives can be evaluated with the previously used binding techniques [24]. Compounds with a ligand efficiency (LE, vide infra) of 0.3 are considered good enough to be optimized to a drug that obeys the Lipinski rules for oral bioavailability [25]. Small fragments having such an LE value might not be detectable by all detection methods which are usually used for FBDD. Therefore, the usage of different and orthogonal methods as well as high testing concentrations of the fragments (µM–mM) are required. For this reason, fluorescence-based competition assays or cell-based assays are usually not suitable for a fragment screening. Such functional assays are used after a successful hit identification to guide hit qualification and optimization. In the following sections, we describe different binding-affinity assays which are used as key technologies for initial hit identification [26,27]. Every technique has its advantages and disadvantages, which are discussed accordingly.

### 3.1. Virtual Screening and Pan-Assay Interference Compounds Filters

As a prescreen, virtual screening (VS), a computer-based method for predicting binding compounds from very large compound libraries to a target protein, can be used. The VS approach allows one to reduce the number of compounds to-be-tested from up to ten million conceivable scaffolds down to a moderate number around one thousand. Virtual hits can be directly tested in different biophysical assays or used to inspire the design of a focused library. Because of the increased accuracy of the computationally programs, virtual screening methods have become more common in FBDD [20]. Several recent reviews cover this topic in detail [28,29,30]. Usually, the VS procedure is designed in a project-specific manner to account for the information available on the target and/or already-known ligands. Often-used approaches involve purely ligand-based pharmacophore modelling and structure-based (target-focused) screenings [31]. Nevertheless, given the considerably smaller size of fragment libraries, VS is usually not required.

Additionally the ligand-based pharmacophore approach, which avoids the need for the macromolecule target structure, has its utility. In this case, a small library of structurally diverse compounds, which are known to interact to a specific target are used to calculate and extract the essential molecular features and functional groups that are important for binding [32,33]. This, of course, requires a significant amount of knowledge about target-interacting compounds to inform the VS campaign, which is not always available.

The target-based approach, also called structure-based virtual screening, starts with docking a defined ligand into a protein target associated with a prediction of the optimal binding mode. Hence, these methods can give an idea how ligands and fragments could bind to a target protein. The advantage of this approach is, that only the X-ray structure of the target protein, e.g., holoenzyme, is needed to conduct the in silico experiment. It should be mentioned, however, that these experiments are hypothetical and have to be followed up with experimental testing [34].

Finally, machine and deep learning principles can be incorporated into the ligand—as can receptor-based VS pipelines, especially when dealing with large data sets [35]. Varieties of these methods have been applied, and the underlying algorithms and concepts range from classical descriptor-driven approaches, e.g., linear quantitative structure-activity relationships (QSAR), to complex, bioinformatics-heavy approaches [35,36]. In the future, it will be interesting to see whether these techniques will hold up to their promise of enabling to design drug molecules from scratch completely in silico [37].

In addition to virtual screening as a prescreening step, so-called pan-assay interference compounds’ (PAINs) can be systematically excluded for further experiments. PAINs often cause false-positive assay results due to unspecific or covalent binding, redox effects, autofluorescence, or degradation. Especially for the selection of a fragment-based library, it is of high importance to identify and eliminate PAINs before applying biophysical screening methods. Obviously, there is an exception to this rule when deliberately using electrophilic fragments for targeting nucleophilic residues in enzymes.

### 3.2. Biophysical Detection Methods for Fragment Screening

#### 3.2.1. Surface Plasmon Resonance

Surface plasmon resonance (SPR) technique is a very powerful tool for the determination of binding events [38]. SPR can be used to measure the binding affinity, specificity, and kinetic parameters of biomolecular interactions between a variety of proteins, DNA/RNA, and small/complex molecules [39,40]. The target protein is first immobilized on a gold or silver sensor surface. Subsequently, a solution of probe flows over the target surface and induces an increase in the refractive index, if binding to the target occurs [41]. In a nutshell, SPR determines changes in reflected light before and after probe–target binding [42]. Dose-response analysis can be applied for determination of dissociation constants *K*_D_ and binding stoichiometry [43,44]. Additionally, the rate constants for association (*k*_on_) and dissociation (*k*_off_) can be determined. Importantly, this methodology is well suited for the detection of very weak probe–target interactions mediated by fragments because of its high sensitivity. The real time monitoring provides the possibility to conduct so-called off-rate screening (ORS), which provides an easy way to observe how long the molecules are interacting with the target. Especially for the FBDD approach, this technique was demonstrated to be very effective for the identification of new fragment hits [45]. The ORS approach provides the possibility to assess the potency of very weak binders and even analyse crude compounds or reaction mixtures by looking at the accompanying dissociation kinetics. Noteworthy compounds with slow off rates possibly have a higher potential to be improved in potency than others [46]. Additionally, the drug residence time of a ligand in complex with its target protein (*t*_R_ = 1/k_off_) can be an important parameter for the functional efficacy of a compound in complex environments such as whole cells [47].

The immobilization of the target on the SPR biosensor can be achieved for a wide range of proteins, including challenging ones, such as transmembrane proteins or G-protein receptors [3,48]. SPR is a label-free technique; hence, the detection of false positives due to fluorescence quenching can be excluded. Additionally, SPR competition assays can be used to gain information on whether a fragment binds competitively to a known substrate or not. Due to the very low protein consumption, the possibility to use SPR in a high-throughput mode, and its cost-efficiency, it is very often used and is effective as a primary selection filter for screenings of large fragment libraries [49,50]. However, direct information about the binding site or interacting groups of a fragment cannot be derived.

#### 3.2.2. Thermal Shift Assay

The thermal shift assay (TSA) is a very reliable and simple technique, which quantifies the denaturation temperature of a protein. The stability of a protein correlates with its melting temperature, which can vary under different conditions; for example, pH-value, buffer composition, amino acid mutations, or binding of a ligand/fragment. Fluorescence-based TSA techniques are the most common ones and are usually referred to as thermofluor assays or differential scanning fluorimetry (DSF) [51]. In principle, the fluorescence of a protein solution is measured in a temperature gradient. An added fluorescence dye (e.g., SYPRO Orange) shows a low fluorescence signal in a polar environment and a high signal in an apolar environment [52]. By denaturation of a protein, the hydrophobic core is exposed, which leads to an increase of the fluorescence signal. Consequently, the melting temperature of the protein can be determined [53,54]. As mentioned before, a ligand–protein interaction can increase or decrease the melting temperature significantly. Hence, a change of this parameter provides important information about protein stabilization or destabilization. In any case, it is now commonly accepted that a change in either direction implies a ligand–protein interaction event [55,56]. Besides the thermofluor variant of TSA, there are also other detection methods for protein stability, such as *N*-[4-(7-diethylamino-4-methyl-3-coumarinyl)phenyl] maleimide (CPM), as a thiol-specific reaction dye; 4-(dicyanovinyl)julolidine (DCVJ), which is rigidity sensitive; or just measuring the intrinsic tryptophan fluorescence lifetime, which differentiates between folded or unfolded proteins [57,58,59]. Non-labelling methods based on detecting changes in the fluorescence of tryptophan are especially easy to use and applicable for all proteins [60,61]. The fluorescence of tryptophan is strongly dependent on the close environment of the protein. A binding event can influence protein folding and/or stability, and by detecting changes in tryptophan fluorescence, the chemical and thermal stability can be determined [60,62]. The nanoDSF technology (Prometheus Series from NanoTemper or nanoDSF from 2bind molecular interactions), e.g., requires only low sample quantities (5 µg/mL), and the measurements are independent from any buffer or detergent. Another similar non-labeling TSA technique for detecting protein stability, called Tycho technology from NanoTemper, relies on protein native fluorescence. It is a simple and rapid technique to check a protein for quality and stability and for analysing ligand–protein binding. An additional advantage of this method is a fast and easy analysis of the quality of a protein during any step of purification, characterization, or assay development.

However, TSA may not be suitable for all target proteins, due to the indirect readout of the melting temperature. Nevertheless, a wide range of applicability, easy and fast handling, and the variable detection methods render this method quite valuable. Due to a high-throughput option and high sensitivity especially, this technique is a useful and powerful method applicable for FBDD as a primary screening step [63]. Still, researchers have to be aware of a usually relatively-high number of false positives. Hence, an orthogonal filtering step is imperative to confirming which compounds are true binders.

#### 3.2.3. Microscale Thermophoresis

Microscale thermophoresis (MST) has been well-established as a technique to detect specific probe–target interactions in recent years. It is a biophysical technique for the characterization of any kind of biomolecular interaction [64,65]. It can be used for proteins, small and complex molecules, fragments, nucleic acids, liposomes, nanoparticles, or ions. In principle, it detects the change in the fluorescence of a labelled target in a temperature gradient as a function of the concentration of a non-fluorescent ligand. The temperature gradient is generated by an IR-laser, and the fluorescence distribution is monitored inside a capillary. This change in fluorescence is based on two main effects. First, the temperature gradient can induce a change in fluorescence. This effect can be influenced by a binding event. Second, thermophoresis of the labelled protein occurs, which is basically the movement of molecules in a temperature gradient. The specific properties of a biomolecule in solution, such as size, surface charge, or hydration shell, influence the thermophoretic profile of the molecule [66]. If a ligand binds to the target, the chemical microenvironment is changed, which leads to a change in the specific thermophoretic profile of the molecule [67]. The detection of the thermophoretic change of a target molecule in relation to varying ligand concentrations can be used to calculate *K*_D_ values. In various studies, it has been demonstrated that MST can also detect weak binders, such as fragments, and is amenable for implementation into HTS campaigns [68,69]. Advantages over other methods are the very low protein consumption and the large number of compounds that can be screened, thanks to short experimentation times [70]. Furthermore, it should be mentioned that the samples are measured directly in solution and there is no need to immobilize proteins, as there is when using, for example, SPR. For MST, mainly, three simple protein labeling techniques are used. These involve crosslinker reactive groups like *N*-hydroxysuccinimide (NHS)-esters for the reaction with primary amines, maleimide functions for labelling sulfhydryl groups, and Tris-NTA for His-tag labelling. Additionally, the shape of the MST traces provides information on protein aggregation or denaturation and hints at specific or unspecific ligand–protein binding during the measurements. Additionally, fluorescence-interfering compounds can directly be identified when analysing fluorescence homogeneity. Using MST can maximize the efficacy of fragment screening campaigns due to the large spectrum of applications and easy handling [69]. All these advantages render this technique applicable and attractive for FBDD [68].

#### 3.2.4. X-ray Methods

Macromolecular X-ray analysis is a key method for FBDD, as it generates very detailed information about the protein–ligand interaction. By this means, it provides the opportunity to conduct structure-based-design studies to improve the ligand affinity efficiently [71,72]. Suitable crystals can be achieved using two different methods, co-crystallization or soaking. To obtain crystals using the co-crystallization method, the ligand is added to the mixture before crystal formation starts. In the soaking technique, the ligand is added directly to a mixture with pre-existing crystals [73]. Both methods can be challenging, since not all proteins are easily crystalized and some ligands can disrupt the crystal lattice. As a prescreening for crystallization, thermal shift assays are often used to determine which compounds stabilize the protein fold, and are therefore favoured. Binding sites, which are occupied by a variety of different fragments, can be identified as “hot-spots” of a given target. By generating various co-crystals, each with a different fragment, the different modes of binding can enable rational fragment linking or merging. Co-crystallization usually is historically not used for a primary screen because it usually requires large amounts (10–50 mg) of highly pure protein [74]. Interestingly, it has become possible to individually screen up to 1000 compounds in less than one week. This, however, requires dedicated instrumentation and highly streamlined processes; e.g., those established by XChem at the Diamond light source or with Astex’s Pyramid Discovery Platform [75].

Astonishingly, it has been demonstrated that even low-affinity binders are detectable by this X-ray technique. Having a co-crystal structure in hand, which resolves the fragment binding site in atomic detail, truly fosters subsequent optimization efforts [76]. However, determining structure–activity relationships is not possible by crystallography alone, because it does not provide direct information about binding affinities.

At Helmholtz Zentrum Berlin (HZB), researchers have developed a workflow for the detection of hit fragments [77]. Firstly, a fragment library is selected which is to be screened for during the campaign. There are several academic libraries that can be chosen from; for example, the F2X-Universal Library, which contains over 1100 compounds, or a sub-selection of 96 fragments called the F2X-Entry Screen. Commercial libraries can be purchased from, for example, Cambridge, Enamin, LiverpoolChiroChem, JBS FragXtal Screen, and MedChemExpress—the MCE Fragment Library. Secondly, co-crystals are formed, and crystallographic data is collected. Thirdly, the data is processed and refined automatically using the programs XDSAPP and PHENIX [78,79]. Lastly, together with the hits after the refinement pipeline, PanDDA analysis allows for the identification of binding ligands from weak signals, which previously would not have been analysable [80].

Another example of a fragment screening platform based on crystallography is FragMAX. This platform follows, in general, the same protocol as described above: firstly, crystallization conditions are optimized; secondly, co-crystals are prepared either via co-crystallization or soaking; and thirdly, data collection and processing lead to hit fragments [81].

#### 3.2.5. Nuclear Magnetic Resonance Methods

The nuclear magnetic resonance (NMR)-based techniques are among the most frequently used methods for analysing ligands and protein–ligand interactions as well as protein structures and dynamics in solution [82]. These are, in principle, very robust techniques and can be used in primary screens for the identification of target binders or to provide deep insights into structural characteristics of specific ligand–protein interactions [83,84]. Advantages of the NMR-based methods are that interactions can be determined on a molecular level and in a non-destructive manner [85]. Additionally, *K*_D_ values can be determined in the µM–mM range. However, NMR-spectrometers are expensive instruments, especially those with large magnets providing more than 700 MHz resolution, and usually need expert support for data processing and analysis. There are three NMR methods, which are most widely used in the FBDD field.

##### Saturation-Transfer Difference NMR

Saturation-transfer difference (STD) NMR is a very simple and powerful technique, especially for detecting weak interactions between ligand and target [86]. It is applicable for initial screens, evaluations of hit or lead structures, and for gaining important information on the binding orientation of a ligand to its target. This ligand-observing method does not require any isotope-labelling of the target protein, nor does it need high protein concentrations (10–50 µM). The technique relies on the dissociation mechanism between ligand and protein (*K_D_* µM–mM). In theory, an STD spectrum (I_STD_) is a difference spectrum between the bound state (on-resonance, I) and the free ligand state (off-resonance, I_o_) spectrum of ligand and protein, *I_STD_* = I − I_o_ [87]. By irradiating only the protein, the saturation is transferred from the protein to the bound ligand. By this means, strongly interacting ligand groups show a higher enhancement than less strongly or non-interacting groups in the STD spectrum. Based on this requirement of proximity, the STD effect can give key information about the orientation of the ligand to the protein. It is also possible to discriminate between specific or non-specific binders [86,88]. Additionally, dissociation constants in the µM–mM range are detectable by performing dose-response experiments [82]. Competition experiments with known binders can also be performed and will provide information whether the fragment in question binds to the same area.

##### Chemical-Shift Perturbation NMR Spectroscopy

The chemical-shift perturbation NMR method is a protein-observing NMR method. It relies on the chemical-shift perturbation of amino acid signals caused by covalent or non-covalent interactions between ligand and protein [89]. For protein detection via NMR, ^15^N and/or ^13^C-protein labelling is required. An NMR spectrum of the labelled protein and one with labelled protein in complex with the ligand have to be measured and compared. Upon ligand–protein binding to a specific site of the protein, the local chemical environment of the amino acids is affected. By this means, a chemical shift of these specific amino acid signals is detectable. The sensitivity of this method brings a lot of advantages, as it can easily detect weak ligand–protein interactions in mM ranges and provides direct information about which amino acids make up the interaction interface [90]. However, protein-observing NMR methods are often time consuming, especially with large proteins because a chemical shift mapping of the single protein is required first. Furthermore, the isotope labelling techniques must be established for the protein beforehand [3].

##### ^19^ Fluorine NMR Spectroscopy

^19^F-NMR spectroscopy can be performed in protein-observing as well as ligand-observing modes. The former method works similarly to the chemical shift perturbation NMR methods using ^15^N or ^13^C-labeled proteins described above and requires the introduction of a fluorinated label into the biomacromolecule, usually via non-natural amino acids [91]. Chemical shifts of ^19^F are extremely sensitive to changes in the local environment induced by protein–ligand interactions or conformational changes of the protein [92]. Compared to ^13^C labelling, ^19^F offers a higher natural abundance and less signal overlap [93]. Additionally, ^19^F has a very broad chemical shift range (500 ppm) and less complexity [94]. The ligand-observed method is used quite regularly, employing specially composed fluorine-rich compound libraries. For this purpose, labelling of the target protein is not required. Interestingly, when using fluorine in both protein and ligand simultaneously, information on the dynamics of the ligand–protein interaction and binding pose can be obtained [95].

#### 3.2.6. Isothermal Titration Calorimetry

Isothermal titration calorimetry (ITC) is a biophysical method to determine thermodynamic parameters of ligand–target interactions in solution [96]. Typically, this method is used to analyze the binding of a ligand or small molecule (e.g., fragment) to a macromolecule (e.g., protein or DNA) [97]. An ITC instrument consists of two cells, a reference cell as a control (containing e.g., buffer) and a sample cell for detecting the specific interaction of interest. The difference in temperature between these two cells is precisely measured. The sample cell contains the macromolecule, and the ligand or small molecule is titrated into the mixture. Form the heat measured, which is released due to the binding event, the enthalpy change (Δ*H*) and the Gibbs free energy (Δ*G*) can be determined. This allows for the calculation of the binding affinity (*K_a_*) and the change in entropy (Δ*S*) [98,99]. Additionally, binding stoichiometry and interactions between more than two molecules can be studied. The discrimination between entropy and enthalpy-driven binders especially, is one of the main advantages. Information about the thermodynamic profile of the binding event can support the understanding of relations between ligand affinities and physicochemical properties [100]. The enthalpic contribution to a binding event can be expressed as the enthalpic efficacy index EE (*EE* = ΔH/Q, *Q* = number of heavy atoms, vide infra). Importantly, fragments mostly bind in an enthalpy-driven manner to the protein surface and often interact at energetically favoured regions—so called hot spots. These hot spots are usually polar residues, which are surrounded by apolar amino acids providing a hydrophobic environment. Highly enthalpy-driven fragments tend to increase the chances of generating high affinity and selective binders [101,102]. Unfortunately for FBDD approaches, this method is often not used as a standard procedure, as fragment interactions are usually too weak for detection [103]. Furthermore, a large amount of protein is required, and ITC is not applicable for high-throughput screening. However, it is particularly beneficial for not only gaining insights in the binding affinity of ligand–target interactions, but the thermodynamic profile of the binding event [104]. Hence, it should be applied as a secondary or tertiary compound evaluation method driving hit prioritization for subsequent optimization [105,106].

#### 3.2.7. Bio-Layer Interferometry

Bio-Layer Interferometry (BLI) is a real-time, label-free (RT-LF) optical technique that allows for monitoring the interaction between an immobilized target on a biosensor surface and a ligand in solution. Binding events can be followed through a shift in wavelength, which is caused by an increase in optical thickness at the surface. It is possible to determine the affinities of small molecules to targets via solution competition experiments [107]. However, it is mostly used to screen for biomolecules; e.g., antibodies [108,109].

## 4. Fragment-Hit Qualification

### 4.1. General Considerations

The methodologies described above should help scientists to identify fragment-sized hits for a given target of interest. In order to embark on a subsequent lead-generation campaign, hits need to be prioritized to be able to focus medicinal-chemistry resources on the most promising starting points. Prioritization of fragment hits is subject to a multi-parameter consideration. Obviously, biological activity is one primary criterion on which to select the starting points for fragment optimization. To this end, effects observed in the orthogonal screening and functional assays should be evaluated together in terms of validated target binding and ligand efficiency (LE) or a related metric to allow a proper comparison. To this end, several quantifiable metrics have been invented to serve as activity-related guideposts for compound selection and optimization. As fragments are of small molecular weight, heavy atom count or a related structure-inherent parameter is usually taken into account to evaluate the quality of the investigated target–ligand interaction. The most straightforward metric in this regard is ligand efficiency (LE). It reflects the ratio between the Gibbs free energy of binding (Δ_B_*G*) and the number non-hydrogen atoms (*N*) (see Equation (1)) [110]. A value of greater than or equal to 0.3 for the LE is considered a suitable starting point for further optimization.
*LE* = (Δ_B_G)/N  with Δ*_B_G* = −RTlnK_i_(1)


It can be simplified to Equation (2) [111].
*LE* = 1.4(p*IC*_50_)/*N*(2)


Several modifications of this equation exist, replacing, for example, the potency term (Δ_B_*G*) either by p*IC*_50_ (as in Equation (2)), the percentage of inhibition, or the enthalpic contribution of the thermodynamic binding profile (enthalpic efficiency, EE; see also ITC section); another is exchanging the size parameter (*N*) by molecular weight or total polar surface area. The EE necessitates the implementation of potentially cumbersome ITC experiments, but is particularly suited for discriminating between enthalpy and entropy-driven binders among the set of hits. The former are considered to form a higher number of oriented, specific, and geometrically well-defined interactions, rendering enthalpy-driven binding highly desirable.

Furthermore, some metrics have been devised taking other compound parameters into account. The ligand-lipophilicity efficiency (LLE) or lipophilic efficiency (LiPE) aims at identifying hits of low hydrophobicity (Equation (3)) [7]. Usually, these compounds can be optimized in an easier fashion, as the enlargement process inherently tends to introduce further lipophilic groups into the compound. Hence, starting from a hydrophilic hit can be advantageous. A special variant of the LLE metric has been devised at Astex Therapeutics, which is referred to as the LLE_AT_ [112]. In that, potency, lipophilicity, and compound size are considered at once (Equation (4)) [7].
*LLE* = p*IC*_50_ or *K*_i_ − clog*P*/*D*(3)
*LLEAT* = 0.111 + [(1.37 × LLE)/*N*](4)


Again a value of LLE_AT_ ≥ 0.3 is considered a suitable starting point for hit-to-lead optimization [112].

In the opinion of the authors, enthalpic efficiency (EE) and LLE_AT_ are particularly useful metrics for the evaluation of fragment hits. The former is, however, not, in any case, a feasible option, as it depends on the availability of a thermodynamic binding profile, which can be difficult to obtain. A variety of additional metrics of potential utility are available for implementation in to the drug discovery pipeline [7].

Additional compound parameters, which should be considered before embarking on an in-depth hit-optimization campaign, are solubility, synthetic tractability, the availability of commercial analogues, and the availability of structural information on the binding mode. The latter can be gathered on the atomic level via X-ray crystallographic analysis of target–ligand complexes or more seldomly through 2D NMR solution structures [113]. This will allow for the analysis of structure-based binding pharmacophores and potentially the comparison to already reported structures. However, especially in early stages of drug discovery projects, such data is often not yet available. Hence, alternative routes to acquiring structural insights into the ligand binding modality can be worth exploring. If at least a holoenzyme structure is available, a few indirect methods become exploitable. For example, combining biophysical methods such as SPR of ITC with single amino acid mutations within the active site of the enzyme can provide essential information about the binding region of the fragment of interest [114]. Additionally, quite often, target enzymes employ reactive residues in order to perform their biological function. The signal molecule synthase PqsD from *Pseudomonas aeruginosa*, for example, utilizes the thiol nucleophile of the Cys112 side chain in order to generate a thioester-bound anthraniloyl reaction intermediate through consumption of anthraniloyl-CoA. This enzyme was previously identified as a suitable anti-biofilm target to tackle *P. aeruginosa* infections [115]. The covalently blocked active site intermediate can be generated on an SPR sensor chip by addition of the anthraniloyl-CoA substrate, and then used for the characterization of fragment hits (Figure 1) [114,116,117]. This enabled researchers, for example, to classify PqsD inhibitor **1** as an active-site binder.

Other possibilities for setting up informative competition experiments can involve, for example STD, NMR, or ITC methodologies. It has to be noted, however, that such approaches only provide low-resolution information of the approximate binding region and usually need to be supported by docking experiments involving inherent ambiguities. Importantly, the more detailed the structural information that is available, the more straightforwardly subsequent optimization steps can be conducted, significantly increasing the chances of success for the fragment-to-lead phase. Hence, the presence of co-crystal structures is highly favoured for fragment-based drug discovery.

### 4.2. Enzyme Targeting

A usual ligand-based approach aims at finding new enzyme inhibitors by deriving them from known structures of their substrates, cofactors, or intermediates. If such a rational design is not successful, the screening tools described above may yield fragment-based hits as alternative starting points [118]. Again, fragments have comparatively weak binding affinities and low functional efficacy. However, the high ligand efficiencies and above-mentioned advantages can pave the way for generating very potent and selective enzyme inhibitors. Evaluating activity in a functional enzyme assay as early as possible in the inhibitor identification and optimization pipeline is an essential cornerstone for a successful medicinal-chemistry campaign. However, in the case of fragment-based drug design, relatively high concentrations of pure, stable, and soluble enzymes are required for reliable assay readouts due to the low affinities of fragments.

X-ray crystallography is also a suitable tool for evaluating the potential of a fragment to inhibit an enzyme after optimization. It is important to have access to robust and high diffraction-quality crystals of the enzyme [76]. Furthermore, the binding site of the enzyme should be devoid of interfering endogenous ligands, so that low affinity fragments are able to bind [119]. Well-defined and deep pockets are often easier to target than shallow grooves [120].

For the identification and characterization of new hot spots for fragment binding, the aforementioned computational screening methods can be beneficial as docking experiments, being able to predict initial structure–activity relationships [121].

### 4.3. Functional Enzyme Assays

For the evaluation of enzymatic inhibitors/optimized fragments and for studying kinetics of the enzymes, several methods are well-established and easily applicable for a wide range of enzymes.

A very sensitive technique to study enzyme inhibition or kinetics is based on mass-spectrometry (MS) detection. Via liquid chromatographic-MS (LC-MS), the specific enzymatic reaction products can be detected and quantified using the reaction mixture aided by LC separation. Benefits of this technique are that it is label-free and that there is no need for substrate modification [122]. Furthermore, it is applicable for most enzymatic reactions and for concentration-dependent enzyme inhibition studies.

In case of targeting proteases, the inhibitory effects of inhibitors are often determined by highly sensitive fluorescence resonance energy transfer (FRET)-based fluorogenic enzyme assays. Typically, a short peptide sequence similar to the sequence for a natural cleavage site of the target enzyme is synthesized and labelled at opposite ends with donor and acceptor/quencher molecules within the FRET distance. The enzymatic activity is determined by monitoring the change in fluorescence intensity. As soon as the labelled substrate molecule is cleaved by the enzyme and the donor–acceptor pair drifts apart, the FRET efficiency drops to zero, leading to an increase in fluorescence [123].

For NADH-dependent enzymes, an enzyme-dependent fluorescence recovery after photobleaching (ED-FRAP) assay can also be used [124]. It relies on the photobleaching of a fluorophore as a product of an enzymatic reaction; in this case, NADH. The oxidative UV photolysis of NADH to NAD^+^ is monitored and correlates with the enzymatic activities. It is applicable for in vitro samples as well as in living cell experiments as an imaging method [125].

Another tool for determining specific enzymatic activities is the usage of fluorogenic and chromogenic substrates. In the latter case, substrates increase or decrease the absorption of light at a specific wavelength by conversion of the substrate to the product by the enzyme. An often-used example for a chromogenic substrate is *p*-nitrophenyl phosphate (pNPP), which is specific for phosphatases. The phosphatases hydrolyse the pNPP to *p*-nitrophenol and inorganic phosphorus leading to a yellow colour of the solution. As a fluorogenic substrate, for example, fluorescein di-phosphate (FDP) can be used. By hydrolysis of FDP by alkaline phosphatases, non-fluorescent FDP is converted into highly fluorescent FDP [126]. Of course, the examples given are just a small selection of available enzyme-assay modalities. In any case, a functional assay should be thoroughly adapted to the targeted system of interest as it will guide the subsequent hit-to-lead optimization phase.

## 5. Fragment-To-Lead Optimization

### 5.1. Growing

The most straightforward approach of turning a fragment-based hit obeying the rule-of-three into a lead-like compound with rule-of-five characteristics is fragment growing. The aim is to install novel target–interaction hot spots by increasing the size of the molecule and attaching additional functional groups. In order to achieve this in a rational design fashion, so-called growth vectors need to be identified. These are essentially positions of the fragment scaffold to which additional atoms, functional groups, or even other dedicated binding scaffolds can be attached. In this regard, the modality of increasing the size of the molecule might play an important role and is characterized by directionality (geometry), substituent size, and the flexibility of the attachment. Actually, identifying these growth vectors might pose a first big hurdle in the process, especially if co-crystal structures are not available. In such scenarios, gaining basic information on inhibitor binding modes, as described above, is key. In the following example, we show how a combination of different biophysical screening and evaluation methods allowed for the identification of the first inhibitors of the protein–DNA interaction between the herpesviral target Latency-associated nuclear antigen (LANA) and its binding sites on the viral episome [127]. Figure 2 shows the screening cascade used for hit identification. First, an SPR screening was conducted, yielding 52 positives. Then, an orthogonal DSF (TSA) filter was applied to preselect 20 target binders. These 20 initial hit compounds were tested for in vitro functionality using a fluorescent polarization (FP)-based competition assay for the quantitative evaluation of target interaction inhibition. In this manner, three very promising fragment hits were prioritized for further optimization and evaluation. In this example fragment **2** has been used for growth vector identification and fragment growing (Figure 2). Fragment **2** showed a low inhibitory effect of 25% at 1 mM in the functional interaction inhibition assay, which is promising, considering the size of the fragment hit.

First, initial analogues were generated to identify a suitable growth vector for subsequent enlargement. That way, it was observed that introducing substituents at the imidazole moiety improves activity. In particular, compound **4** (Figure 2) could improve the inhibitory effect to 91% at 1 mM. Additionally, different functional groups at the phenyl ring instead of the NH_2_-group were tested, showing that a carboxylic acid (compound **3**) improves activity to an IC_50_ value of 333 µM. These first derivatisation results of initial hit **2** led to further optimization studies with a combichem approach using click chemistry. Replacing the imidazole moiety with a triazole core, keeping the carboxylic acid function at the phenyl ring, and increasing the size of the molecule at the Eastern part led to a small compound library of 29 new derivatives. This procedure facilitated the discovery of compound **5** by the introduction of a bulky pyridine ring. Still possessing a fragment-like size, this compound represents a promising inhibitory activity (*IC*_50_ = 17 µM). Additionally, it shows favourable physicochemical properties (low clogP of 2.00, good solubility) and suitable ligand efficiency (*LE* = 0.33). Hence, it is a promising starting point for a follow-up lead-generation campaign. As this example demonstrates, fragment growing and, in general, compound enlargement can be attempted with and without structural information. Although, for obvious reasons, the latter usually requires more effort and is less likely to be successful.

### 5.2. Merging

If two ligands or fragments bind to overlapping regions of the target protein, motifs of both can be merged into one compound. Ideally, the favourable affinity-mediating features of both compounds work additively or even synergistically when fused into one unit. Fragment merging can be attempted when obvious structural similarities of the two ligands in question are present. For example, if it is plausible to assume that both share a certain prominent binding motif or coordinate to an active-site transition metal, but do otherwise not align well, they might diverge into different regions of the active site. The geometry and attachment patterns of the combined binding groups is key for productive cooperative binding. Hence, several possible merging opportunities should be explored to identify the optimal shape of the resulting merged compound. Here, and especially in cases where the to-be-merged compounds share no obvious structural similarity, structural data of ligand–target complexes are necessary to guide this approach. Again, X-ray co-crystal structures as well as NMR solution structures deliver the most reliable and detailed source of information. Nevertheless, wet-lab experiment-guided docking studies may also provide insights robust enough to justify the syntheses of merged compounds. One such example is provided below. In previous studies, two distinct inhibitor classes of the bacterial target enzyme PqsD were identified (Figure 3) [114,128]. Docking as well as SPR competition experiments suggested that fragment **1** (vide supra) and compound **6** had overlapping binding sites and shared a common phenylmethanol feature. Hence, compound **7** was synthesized to merge both units into one scaffold [129]. Unfortunately, the resulting fused molecule did not show better potency on the target enzyme than the frontrunner compound **6**. Nevertheless, the substantial modifications applied to the inhibitor were tolerated and brought about respectable activity. This example nicely illustrates the general strategy and emphasises the need for structural information, which has to be acquired beforehand.

### 5.3. Linking

For a fragment linking approach, two or more fragments have to bind to different but adjacent sites of the enzyme active site [3]. This approach is similar to the fragment merging strategy laid out above. However, it introduces one additional component into the ligand system: a linker moiety. Finding the right linker motif, which orients the individual fragment units in the favourable geometry in relation to each other without introducing too much flexibility whilst maintaining the binding poses of both fragments, can be very challenging. If successful, the combination of two fragments with rather low affinity could result in significantly higher affinity and has the potential to result in “superadditive” contributions of both binding motifs. The challenge in fragment linking is the exploration of the binding mode of both fragments and the identification of an optimal linker. Only in this case, the overall reduced so-called rigid body entropy translates into synergistically-improved affinity. By binding of a fragment to a target protein, rotational and translational entropy are lost. This entropy penalty has to be overcompensated by attractive interactions formed between the ligand and the target. When two fragments bind in parallel to adjacent sites, each has to pay this entropy penalty. When these two fragments are linked together in an ideal way, the resulting singular compound only encounters the loss of rigid body entropy once. Hence, the affinity observed will be much greater than only the sum of the individual affinities [130]. The additional binding energy gained is often also referred to as linker energy. To overcome the challenges associated with fragment linking, we pioneered a synergistic combination with dynamic combinatorial chemistry (DCC). For this proof-of-concept study, we used the model enzyme endothiapepsin [131]. Generally, such aspartic proteases are found in fungi, vertebrates, plants, and retroviruses such as HIV. The class of enzymes plays a causative role in diseases such as malaria, Alzheimer’s disease, hypertension, and AIDS. X-ray crystal structures of endothiapepsin in complex with fragment inhibitors **8** and **9** (PDB IDs: 4KUP and3T7P) identified by DCC were used as a starting point for fragment linking studies facilitated by DCC. Hits **8** and **9** displayed *IC*_50_ values of 12.8 µM and 14.5 µM and LEs of 0.27 and 0.29, respectively. These hit compounds are not typical small fragments. However, as was previously mentioned, also structures which do not follow the RO3 principles can be used in FBDD approaches. After superimposing the binding modes of the two hit structures, it was envisioned that these moieties could possibly be linked, since they occupy different pockets in the protein. The linking of **8** and **9** should, therefore, generate an inhibitor that occupies multiple binding pockets of endothiapepsin (Figure 4).

Using molecular modelling, an acylhydrazone motif was selected as a suitable linker structure. The acylhydrazone linker provides H-bond donor and acceptor sites, which could enhance the binding affinity. Acylhydrazones can be formed in a reversible fashion from the condensation reaction between hydrazides and aldehydes. Reversibility is key for DCC, as it allows an external stimulus, such as a protein target, to influence the equilibrium. In the so called target-directed DCC, the target can stabilize certain members of the dynamic combinatorial library (DCL), giving rise to an amplification of ligands with high affinity at the cost of other DCL members (Figure 5). In this way, a moderate number of molecules can be screened for, without having to synthesize, purify, and analyse each molecule separately [132,133].

The homo-bis-acylhydrazones **10** and **11** were hits from the DCC experiments and were synthesized and evaluated accordingly. Compared to compound **9**, the potency of inhibitor **10** was increased 240-fold, yielding an *IC*_50_ value of 0.054 µM and a LE value of 0.29. For inhibitor **11** an *IC*_50_ of 2.1 µM and a LE value of 0.25 was determined (Figure 4) [106]. Obviously, only the symmetric linking modality resulted in efficient cooperative binding.

## 6. Conclusions

FBDD has become a respectable alternative to high-throughput screening of large compound libraries. Due to the moderate instrumentation requirements as well as general cost-effectiveness, this approach is attractive for both the academic and industrial drug discovery settings. However, as laid out in this review, expertise from fields of organic chemistry, biophysical screening, structural biology, compound profiling, and hit qualification have to work together right from the beginning to facilitate successful fragment-based drug discovery. Hence, integrated medicinal chemistry teams have to be established on site in order to tackle the challenges occurring en route to lead-like compounds or even preclinical candidates. The importance of structural data and an in-depth qualification of fragment hits cannot be emphasized enough in order to enable rationally guided fragment growing, merging, or linking trials. If all ends meet up, FBDD represents a facile, target-driven strategy for finding suitable starting points ideal for lead generation circumventing HTS.

## Figures and Tables

**Figure 1 molecules-24-04309-f001:**
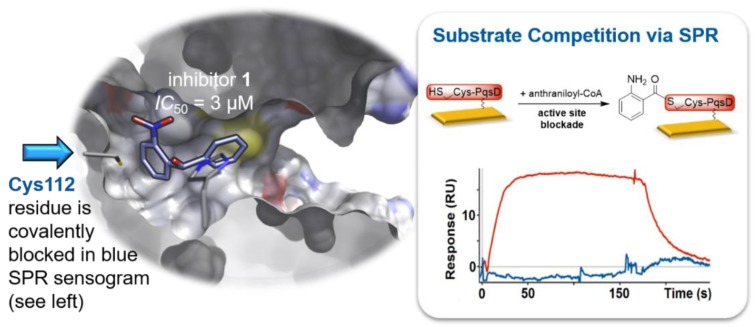
Schematic depiction illustrating the SPR-based competition experiment. (Left) The docking mode of inhibitor **1** bound to the target enzyme PqsD and the active site cysteine (Cys112) are indicated. (Right) SPR sensorgrams with (blue) and without (red) covalent active-site blockade via substrate preconditioning (*RU* = SPR response units) are given. The lack of SPR response (blue) indicates that inhibitor **1** is binding at the active site [114].

**Figure 2 molecules-24-04309-f002:**
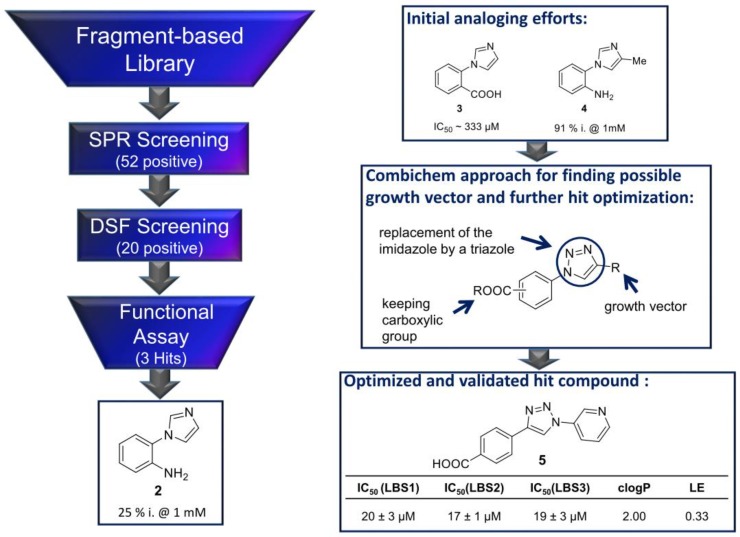
Fragment screening cascade using: SPR and DSF selection steps, followed by functional evaluation via fluorescence polarization, which resulted in three fragment hits. Growth vector identification through initial analoging allowed for growing of fragment **2** into a double-digit micromolar validated hit.

**Figure 3 molecules-24-04309-f003:**
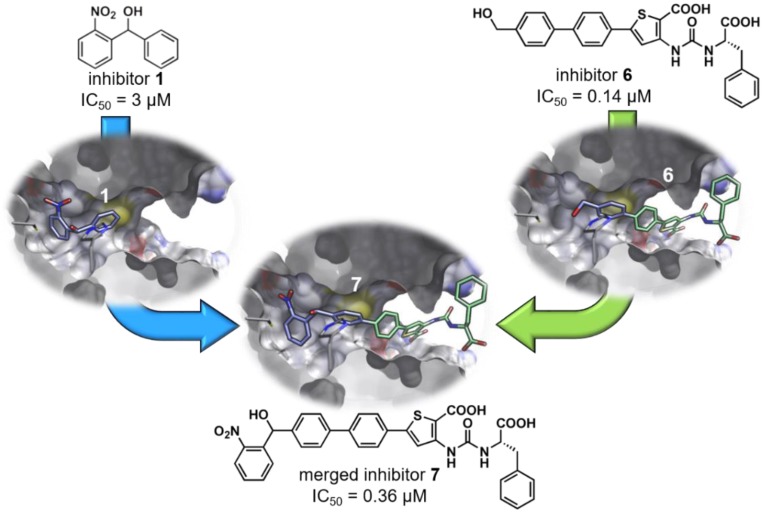
An example of a compound merging attempt. Fragment **1** and compound **6** were merged, guided by an SPR-informed docking poses, yielding inhibitor **7**. If X-ray structures for the individual inhibitors were available, this attempt could be improved by adjusting the merging modality [129]. Light blue: carbon atoms of inhibitor **1** and carbons resembling an overlapping motif in inhibitor **6**. Light green: the other carbon atoms of inhibitor 6. Blue: nitrogen. Red: oxygen. Hydrogen omitted for clarity.

**Figure 4 molecules-24-04309-f004:**
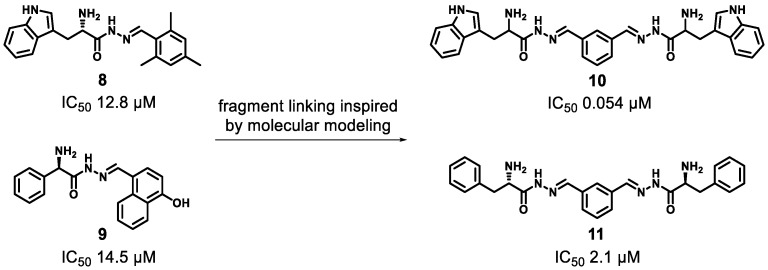
Structures of hits **8** and **9** and linked bisacylhydrazone linked inhibitors **10** and **11** [131].

**Figure 5 molecules-24-04309-f005:**
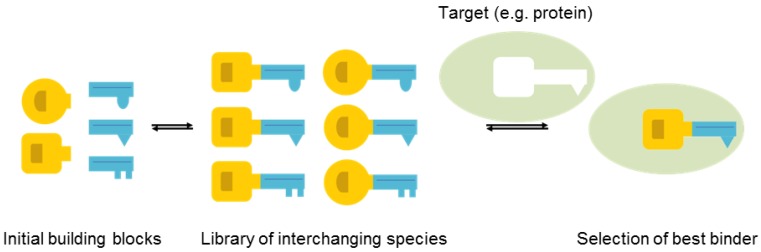
Schematic representation of target-directed dynamic combinatorial chemistry.

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
