# Peer review of "Concepts and Core Principles of Fragment-Based Drug Design"

_molecules, 2019, doi:10.3390/molecules24234309_

Round 1
Reviewer 1 Report
The manuscript by Kirsch and colleagues describes basic concepts of Fragment-based drug design. FBDB is currently one of the method of choice in drug discovery both in Academia and Industry, thus I find this review article very timely and important. The manuscript is well-written and very informative. The authors managed to comprehensively review principles of FBDD including the description of main methodology used in the FBDD pipeline. I recommend the manuscript by Kirsch et al for publishing in Molecules journal after minor revisions.
Minor revisions
Line 86 – Lipinski rules are not described properly along with the rule of three. The non-specialist reader could be mislead if Lipinski rule is rule of three or rule of five or something else. Please be more precise. Line 103 – orthogonal methods. Please provide the meaning to the reader, ie. it is a method of a very different principles Line 104 – remove double-space after “enables” Line 108 – “mM” instead of “mm” Line 109 – particularly 121 – in the following what? Add object eg. “paragraphs” Par 3.2.2. TSA – please discuss other methods that do not require labels e.g. Prometheus or Tycho from Nanotemper 2.4 – I find this paragraph very short in comparison to NMR methods. I would be more descriptive. You may add information about PanDDA software, Acoustic Tranfer Systems, FragMax pipeline, commercial and academic fragment libraries for crystallization (eg. BESSY library) . 2.5 – 3.2.8 – should be combined in one paragraph 3.2.5 NMR methods with sub-paragraphs 3.2.5.1… Line 306 – using fluorine in protein? What does it mean? That proteins have Fluorine-label? Be more clear. Line 333 – I miss paragraph about other methods e.g. Octet.. Line 451 – remove double-space between “For” and “NADH-dependent” Line 455 – citation missing for ED-FRAP
Author Response
Reviewer1:
Line 86 – Lipinski rules are not described properly along with the rule of three. The non-specialist reader could be mislead if Lipinski rule is rule of three or rule of five or something else. Please be more precise.
We thank the reviewer for this comment. We added the following sentences and modified the paragraph slightly for the sake of clarity:
“Furthermore, a number of rotatable bonds (NROT) ≤ 3 and a polar surface area (PSA) ≤ 60 can be beneficial.[18] The rule-of-three, is closely related to the prominent Lipinski´s rule of five (RO5), which makes use of the same molecular descriptors but using a less restricted value of five as the cutoff (molecular weight ≤ 500 Da, number of hydrogen bond donors ≤ 5, number of hydrogen bond acceptors ≤ 10, logP ≤ 5). Importantly, the latter was derived to provide a predictor for the oral bioavailability of the drug candidate at hand, while the former is used to evaluate fragments regarding their suitability to be optimized into RO5-obeying drug candidates. RO3 as well as the rule-of-fiveR05, should be considered as rough guidelines in drug design for achieving orally bioavailable compounds.”
Line 103 – orthogonal methods. Please provide the meaning to the reader, ie. it is a method of a very different principles
We agree with the referee that a little bit more information makes it easier to follow for the reader. We have clarified this part by modifying the section in the following way:
Line 115 ”There are some tasks necessary for a successful FBDD campaign, like carefully selecting the fragment library, applying several orthogonal methods to confirm that the fragment is binding to the target and characterizing of the fragment’s binding mode. In this regard, orthogonal screening methods refer to e.g. biophysical or biochemical techniques, which interrogate the ligand-target binding but relying on different measurement principles”
Line 104 – remove double-space after “enables”
We have removed double-spaces throughout the manuscript.
Line 108 – “mM” instead of “mm”
We have carefully checked and changed this throughout the manuscript.
Line 109 – particularly 121 – in the following what? Add object eg. “paragraphs”
Indeed, the word ‘sections’ was missing here. We have added the subject accordingly.
Par 3.2.2. TSA – please discuss other methods that do not require labels e.g. Prometheus or Tycho from Nanotemper
We agree with the reviewer that these non-labeling TSA/DSF techniques are also important to discuss as these techniques are very easy, fast and widely applicable. We have added a few sentences in the TSA chapter to improve our manuscript and give more information to these methods for the reader.
Line 223 “…which differs between folded or unfolded proteins.[57–59] Especially non-labeling methods based on detecting changes in the fluorescence of tryptophan are easy to use and applicable for all proteins.[60,61] The fluorescence of tryptophan is strongly dependent on the close environment of the protein. A binding event can influence protein folding and/or stability and by detecting changes in tryptophan fluorescence, the chemical and thermal stability can be determined.[60,62] The nanoDSF technology (Prometheus Series from NanoTemper or nanoDSF from 2bind molecular interactions) e.g. requires only low sample quantities (5 µg/mL) and the measurements are independent from any buffer or detergent. Another similar non-labeling TSA technique for detecting protein stability, called Tycho technology from NanoTemper, relies on protein native fluorescence. It is a simple and rapid technique to check a protein for quality and stability or analyzing ligand-protein binding. An additional advantage of this method is a fast and easy analysis of the quality of a protein during any step of purification, characterization or assay development.
However, TSA may….”
3.2.4 – I find this paragraph very short in comparison to NMR methods. I would be more descriptive. You may add information about PanDDA software, Acoustic Tranfer Systems, FragMax pipeline, commercial and academic fragment libraries for crystallization (eg. BESSY library) .
We have included two examples of a fragment screening workflow, based on X-ray. These examples mention the computer programs that are applied in the protocol. We have provided references to these programs.
Line 293 “….information about binding affinities.
At the Helmholtz Zentrum Berlin (HZB), researchers have developed a workflow for the detection of hit fragments.[77] Firstly, a fragment library is selected which will be screened for during the campaign. There are several academic libraries that can be chosen from, for example, the F2X-Universal Library, which contains over 1100 compounds, or a sub-selection of 96 fragments called the F2X-Entry Screen. Commercial libraries can be purchased from for example: Cambridge, Enamin, LiverpoolChiroChem, JBS FragXtal Screen and the MCE Fragment Library from MedChemExpress. Secondly, co-crystals are formed and crystallographic data is collected. Thirdly, the data is processed and refined automatically using the programs XDSAPP and PHENIX.[78,79] Lastly, together with the hits after the refinement pipeline, PanDDA analysis allows for the identification of binding ligands from weak signals, which previously would not have been analyzable.[80]
Another example of a fragment screening platform based on crystallography is FragMAX. This platform follows in general the same protocol as described above: firstly, crystallization conditions are optimized, secondly, co-crystals are prepared either via co-crystallization or soaking and thirdly, data collection and processing lead to hit fragments.[81]”
3.2.5 – 3.2.8 – should be combined in one paragraph 3.2.5 NMR methods with sub-paragraphs 3.2.5.1…
We have changed the paragraph numbering, and combined the NMR methods under paragraph 3.2.5. as was suggested by the referee.
Line 306 – using fluorine in protein? What does it mean? That proteins have Fluorine-label? Be more clear.
The use of fluorine offers the possibility to screen libraries, where the fragments are fluorinated and not the protein. We have made this clearer in the text by modifying the section:
Line 356 “19F-NMR spectroscopy can be performed in protein-observed as well as ligand-observed mode. The former method works similar to the chemical shift perturbation NMR methods using 15N- or 13C-labeled proteins described above and requires the introduction of a fluorinated label into the biomacromolecule usually via non-natural amino acids.[91] Chemical shifts of 19F are extremely sensitive to changes in the local environment induced by protein–ligand interactions or conformational changes of the protein.[92] Compared to 13C labeling, 19F offers a higher natural abundance and less signal overlap.[93] Additionally, 19F has a very broad chemical shift range (500 ppm) and less complexity.[94] The ligand-observed method is used quite regularly employing specially composed fluorine-rich compound libraries. For this purpose, labeling of the target protein is not required. Interestingly, when using fluorine in both, protein and ligand simultaneously, information on the dynamics of the ligand–protein interaction and binding pose can be obtained.[95]”
Line 333 – I miss paragraph about other methods e.g. Octet.
We have added a new paragraph about bio-layer interferometry. In our opinion, this method is not often used for fragments but rather to analyze macromolecule-macromolecule interactions:
“3.2.7. Bio-Layer Interferometry
Bio-Layer Interferometry (BLI) is a real-time, label-free (RT-LF) optical technique that allows for monitoring the interaction between an immobilized target on a biosensor surface and a ligand in solution. Binding events can be followed through a shift in wavelength, which is caused by an increase in optical thickness at the surface. It is possible to determine the affinities of small molecules to targets via solution competition experiments.[107] However, it is mostly used to screen for biomolecules like e.g. antibodies.[108,109]”
Line 451 – remove double-space between “For” and “NADH-dependent”
We have checked this, but there is no double-space. The text is ‘justified’, and in this case, it gives a bit more space between the words than at other places.
Line 455 – citation missing for ED-FRAP
We have added the reference, which was missing.
Reviewer 2 Report
This review provided a general introduction to fragment-based drug design and the underlying concepts. But there are some problems should be modified:
Multiple criteria should be referenced to make the molecule library, so the current commonly applied strategies should be summarized in detail in the 2.1. General Principles for Library Design. There are many approaches of VS, such as based on receptor, based on ligands, or deep learning, thus, the advantages and disadvantages should be compared. These principles can be generally applied to most classes of drug targets. So the application and non-application aspects should be described in detail. In Figure 1 “The lack of SPR response (blue) indicates active site 407 binding of inhibitor 1.[80.]” And Figure 4 “Structures of hits 8 and 9 and linked bisacylhydrazone linked inhibitors 10 and 11.89.” The format of the reference is different. It is recommended that the format be read carefully and modified. There are too few references in the past 5 years. The most recent important research paper should be cited. The figure (for example, the color scheme of Figure 3 and 1 are not clear) and grammar should be improved.
Author Response
Reviewer 2:
Multiple criteria should be referenced to make the molecule library, so the current commonly applied strategies should be summarized in detail in the 2.1. General Principles for Library Design.
Indeed, the library design and the question how to design a library are very important. In order to improve our manuscript further we added a part of applied strategies for making a fragment library:
Line 67- “…library for matrix-metalloproteinases (MMPs). To assemble a project-focused fragment library with a high chemical and structural diversity some criteria which are important to consider. The outcome of an FBDD project is directly influenced by the composition of the applied library.[10] Firstly, commercially available fragments and fragment libraries should be analyzed. It is important to use a library that meets some primary criteria depending on the profile of the respective target.[11] Usually, commercially available fragment libraries have been selected based on chemical and size diversity and different, well-balanced properties to cover most of the important features. The overall diversity of the library can also be improved by using a pharmacophore-based selection. Secondly, a set of natural products or natural-product-inspired fragments are often useful and could be included.[12] Additionally, for future plans it is important to identify a series of non-commercially available fragments, which came from synthetic chemistry efforts for example from an in-house library or collaborating groups. Such scaffolds can provide a basis for future medicinal chemistry optimization strategies.[13,14]
However, it must be noted….”
There are many approaches of VS, such as based on receptor, based on ligands, or deep learning, thus, the advantages and disadvantages should be compared. These principles can be generally applied to most classes of drug targets. So the application and non-application aspects should be described in detail.
Indeed, methods of VS are plentiful and are ideally modified and adopted to the project-specific screening problem at hand and a number of detailed reviews are available on the topic. The receptor- and ligand-based methods have already been mentioned in our manuscript (section 3.1) Hence, we added following paragraphs to this section spotlighting the points mentioned by the reviewer and referencing suitable subject-focused reviews for the interested reader.
Line 144-“…FBDD.[20] Several recent reviews cover this topic in detail.[28–30] Usually, the VS procedure is designed in a project-specific manner to account for the information available on the target and/or already known ligands. Often used approaches…”
and
Line 160- “...experimental testing.[34]
“Finally, machine and deep learning principles can be incorporated into the ligand- as well as receptor-based VS pipelines especially when dealing with large data sets.[35] Various of these methods have been applied and the underlying algorithms and concepts range from classical descriptor-driven approaches, like e.g. linear quantitative structure-activity relationships (QSAR), to complex bioinformatics-heavy approaches.[35,36] In the future, it will be interesting to see, whether these techniques will hold up to their promise of enabling to design drug molecules from scratch completely in silico.[37]
In addition to…”
In Figure 1 “The lack of SPR response (blue) indicates active site 407 binding of inhibitor 1.[80.]” And Figure 4 “Structures of hits 8 and 9 and linked bisacylhydrazone linked inhibitors 10 and 11.89.” The format of the reference is different. It is recommended that the format be read carefully and modified.
We have changed it accordingly.
There are too few references in the past 5 years. The most recent important research paper should be cited.
We have added more references from the past years. According to the updated reference list these have the numbers [9-15] [28-30], [35-37], [40], [42] [44], [57-62], [65], [77-81], [84], [91], [100], [101], [107-109], [124-126]
The figure (for example, the color scheme of Figure 3 and 1 are not clear) and grammar should be improved.
We have checked to the captions of the figures and improved the grammar where necessary in the manuscript.
Round 2
Reviewer 2 Report
The authors have addressed all my concerns, and the paper is ready for publication in current form.